# Updating the Landscape for Functioning Gonadotroph Tumors

**DOI:** 10.3390/medicina58081071

**Published:** 2022-08-08

**Authors:** Georgia Ntali, Cristina Capatina

**Affiliations:** 1Department of Endocrinology, Diabetes and Metabolism, Evangelismos Hospital, 10676 Athens, Greece; 2Department of Endocrinology, Carol Davila UMPh, 011863 Bucharest, Romania; 3Department of Pituitary and Neuroendocrine diseases, CI Parhon National Institute of Endocrinology, 011863 Bucharest, Romania

**Keywords:** functioning, gonadotroph pituitary adenoma, ovarian hyperstimulation syndrome, precocious puberty, testicular enlargement

## Abstract

Functioning gonadotroph adenomas (FGAs) are rare tumors, as the overwhelming majority of gonadotroph tumors are clinically silent. Literature is based on case reports and small case series. Gonadotroph tumors are poorly differentiated and produce and secrete hormones inefficiently, but in exceptional cases, they cause clinical syndromes due to hypersecretion of intact gonadotropins. The clinical spectrum of endocrine dysfunction includes an exaggerated response of ovaries characterized as ovarian hyperstimulation syndrome (OHSS) in premenopausal females and adolescent girls, testicular enlargement in males, and isosexual precocious puberty in children. Transsphenoidal surgery and removal of tumor reduces hormonal hypersecretion, improves endocrine dysfunction, and provides tissue for further analysis. Medical therapies (somatostatin analogues, dopamine agonists, GnRH agonists/antagonists) are partially or totally ineffective in many cases, especially with respect to antitumor effect. This review aims to update recent literature on these rare functioning tumors and highlight their therapeutic management.

## 1. Introduction

Gonadotroph adenomas account approximately for 40% of all pituitary adenomas [1]. Based on the WHO 2017 classification, they belong to the (steroidogenesis factor 1) SF-1 lineage and are immunoreactive for follicle-stimulating hormone (FSH), luteinizing hormone (LH) β-subunits, alpha subunit (α-SU), steroidogenesis factor (SF-1), estrogen receptor-alpha (ERα), and GATA-3 expression [2].

They are usually hormonally silent tumors, but in rare cases, they provoke clinical syndromes related to gonadotropin secretion. These exceptional clinical manifestations include gonadal disturbances and in particular menstrual disorders and ovarian hyperstimulation syndrome (OHSS) in premenopausal females and adolescent girls, testicular enlargement in males, and isosexual precocious puberty in children [3].

Functioning gonadotroph adenomas (FGAs) are rare, and their exact prevalence is unknown. Interestingly, in a retrospective study of 171 premenopausal women operated for a macroadenoma, Caretto et al. detected three cases with clinical and biochemical characteristics of OHSS and estimated that the prevalence could reach 8.1% in the gonadotroph adenoma group and 2.9% in the nonfunctioning adenomas (NFAs) group [4]. Nevertheless, this has not been validated in other series and could represent an overestimation. The purpose of this review is to update recent literature on FGAs and focus on their therapeutic management.

## 2. Methodology

We performed a narrative review based on relevant articles written in English from a *PubMed* search using the following research terms: “functioning gonadotroph adenoma”, “ovarian hyperstimulation syndrome”, “testicular enlargement”, and “ precocious puberty”. Most papers were case reports or small case series.

## 3. Pathogenesis-Pathology

Only hypothesis have been made about this exceptional behavior of gonadotroph adenomas. Alteration of the pulsatile secretion of gonadotropins and a slight but constant release of FSH could stimulate recruitment of multiple dominant follicles and release of high serum estradiol similarly to ovarian stimulation with exogenous FSH when administered for fertility treatment. Modifications of their heterodimer of alpha and beta chain can potentially lead to an increased bioactivity of FSH levels. Pigny et al. demonstrated with chromatofocusing analysis that a functioning gonadotroph adenoma secreted mostly basic FSH isoforms (pH > 6) in comparison with FSH isoforms from normal pituitary and non-functioning gonadotroph adenomas, which were detected at pH < 5.5. The patient had increased FSH levels 29 (1–7 IU/L), low LH 1 (0.5–10 IU/L), low testosterone levels 3.35 (3–10 ng/mL), and testicular enlargement, implying that these specific FSH isoforms could trigger testicular growth [5]. The degree of glycosylation of the gonadotropins also affects their bioactivity, so secretion of glycosylated variants with increased biological activity may contribute [5,6,7]. FSH bioactivity was investigated through a Sertoli cell aromatase bioassay in 9 patients with gonadotropin secreting tumor, in 21 with nonfunctioning (NFPA), and 30 normal subjects. Male patients with gonadotropin secreting tumors showed a high FSH bioactivity in contrast to women, and interestingly, a boy with precocious puberty had the highest value. Among the NFPA patients, one man secreted a highly bioactive FSH [8]. Increased FSH bioactivity (above 10 IU/L) and a bioactivity/immunoreactivity ration of 1.2–1.3 has been estimated in a young female with FSH-secreting adenoma and OHSS [9]. On the other hand, Kajitani et al. reported a patient with a pituitary macroadenoma, enlarged multicystic ovaries, and normal FSH levels, whose bioactivity did not differ in comparison with that of normal subjects [10].

Mutations of FSH receptor have been associated with spontaneous ovarian hyperstimulation syndrome, but until now, this has not been demonstrated for cases with FGAs [11,12]. Kottler et al. demonstrated that gonadotropin releasing hormone receptor *(GnRHR)* gene was expressed more often in tumors with inappropriate or high gonadotropin levels (functioning) rather than in nonfunctioning gonadotroph adenomas although it was absent in the only case with normal gonadotropins and very high estradiol levels. No pathogenic mutations were determined [13]. A change of tumor functionality from a nonfunctioning gonadotroph adenoma to a functional one causing hyperestrogenemia has been described [14].

Endocrine active gonadotroph adenomas exhibit a strong nuclear staining with Steroidogenesis Factor-1 (SF-1) transcription factor and variable intensity of immunoreactivity for (α-SU), β-FSH, and β-LH [9,15,16,17]. LH immunopositive-only adenomas are rare [18,19]. Eisenberg et al. reported the case of a female with clinical and laboratory findings consistent with a FGA; three pituitary adenomas that stained differently were detected; one stained positively for Steroidogenesis factor-1 (SF-1), Pituitary Transcription factor 1 (PiT-1), Follicle Stimulating Hormone (FSH), and Luteinizing Hormone (LH); a second for SF-1 and PiT-1; and a third for PiT-1 [20]. Wang et al. demonstrated that Kisspeptin 1(Kiss1) mRNA levels were significantly higher in the group with endocrine active gonadotroph adenomas than in the nonfunctioning gonadotroph (NFGA) group and correlated with the preoperative serum estradiol levels suggesting a potential role for the Kisspeptin 1 (Kiss1)/Kisspeptin 1 Receptor (Kiss1R), system in the secretion function of FGAs [21].

## 4. Females

### 4.1. Clinical Findings

Clinical manifestations in premenopausal females are not specific, and they include a variety of menstrual disorders (secondary amenorrhea, oligomenorrhea, severe menorrhagia) [22,23], infertility [24], and in a few cases, the distinct syndrome of ovarian hyperstimulation [18,25,26,27]. This syndrome may present in mild form or as an emergency with abdominal pain and distension if ovarian cysts twist [28,29].

Wang et al. reviewed data of 65 premenopausal patients with FGAs. Sixty patients (92%) had elevated FSH levels, and 7.7% had increased LH. Elevated FSH was associated with menstrual disorders (in 86.7%), infertility (in 16.7%), and multicystic ovaries in 98.2% (55/56). None of the five patients with increased LH demonstrated ovarian enlargement or hyperestrogenemia. Macroadenomas accounted for 89.2% [21]. Hasegawa et al. reviewed data from 50 patients and estimated that the mean maximal adenoma diameter was 22 mm, while 10 patients had Knosp grade ≥ 3 [30]. A unique case in which spontaneous pregnancy exacerbated OHSS in a patient with FGA with ascites and thrombophlebitis highlights the role of hCG in promoting vascular endothelial growth factor expression and vascular permeability [31].

Ultrasound often shows multifollicular ovarian enlargement [22] similar to the iatrogenic complication induced by in vitro fertilization treatment [32]. They differ from polycystic ovary syndrome (PCOS), which is characterized by the presence of 12 or more follicles in each ovary measuring 2–9 mm in diameter.

In postmenopausal women, clinical findings are similar to a nonfunctioning macroadenoma, as the ovaries do not respond to FSH stimulation. As gonadotropin increase is secondary to menopause, low levels of LH levels or discrepancy of FSH and LH could be an indication of a gonadotroph adenoma but not specifically of a functioning one.

### 4.2. Hormonal Profile

Hormonal profile is variable with either elevated FSH and LH levels [30] or normal/high levels FSH and low or low-normal LH and either elevated or not levels of gonadal steroids [9,16,18,19,20,21,22,23,24,27,28,29,30]. Serum LH is often suppressed despite LH positivity in tumor immunostaining, possibly associated to a negative feedback or an irregular LH secretion. Hyperestrogenism is a predominant biochemical finding [18,33,34,35,36,37], and estrogen levels may range from marginally [38,39] to markedly elevated [16,32]. Normal [9,34,37] or elevated [35,40] levels of α-SU and inhibin [35,36,38] have been reported. Prolactin levels may increase due to the stalk effect of a macroadenoma. Paradoxical response after Thyrotropin Releasing Hormone (TRH) stimulation is not always observed [9].

## 5. Males

In males, the case reports of functioning pituitary adenomas secreting gonadotropins, mainly FSH, are very scarce.

### 5.1. Clinical Findings

The overwhelming majority of FGAs in men are macroadenomas, most with supra- and/ or parasellar extension [41,42,43]. The only published case of a presumed FSH-secreting microadenoma in an adolescent male was not confirmed pathologically [44].

Usually, FSH is secreted in excess, and the LH and testosterone levels are normal or decreased [45,46]. Bilaterally enlarged testes due to stimulation of the seminiferous tubules by FSH characterize a few cases [42,45]. The degree of testicular enlargement is variable and size even over 30 mL has been reported [42,43,47]. Numerous factors are responsible for that heterogeneity: very slow effect of FSH, variable biological activity of the FSH, variable duration from onset to diagnosis, [3,47].

The hypothesis of the lack of biological activity in some cases with increased gonadotropin secretion is supported by evidence from case reports. For instance, in a 55-year-old male patient with a giant pituitary adenoma (PA) with very high serum FSH levels, neither testicular enlargement nor increased sperm count were present. In fact, small testicles were present despite significantly increased serum FSH (72.5 mIU/mL). Normal-size testicles were present in a male patient with serum FSH of 106.6 mIU/mL [48]. The lack of the synergic effect of testosterone (in cases with hypogonadism) could also contribute.

In patients with elevated FSH levels, enlarged testes may differentiate a gonadotropinoma from primary gonadal failure. Increased inhibin levels may be a useful adjuvant biochemical marker for these tumors [45]. In all cases, the increase in testicular size should be differentiated from other possible causes (see [3]).

Sexual dysfunction is variably reported: in six out of six males in one series (not all with low testosterone levels) [46], while in another series, less than half of the patients reported it [41]. Not all cases with documented hypogonadism have clinical expression [45,49]. In very large tumors with minimal tumoral secretion of FSH only, compression symptoms can be present [50].

In rare cases, biologically active LH causes increased testosterone concentration. LH tumors are very rare, found exclusively in men in one series [17], and are not associated with a clinical syndrome [15,51,52]. In two cases, increased testosterone induced polycythemia, which reversed after hormonal control [48,53] and in one case, gynecomastia due to peripheral conversion of androgens to estradiol [53].

### 5.2. Hormonal Concentrations

In most cases, FSH serum concentrations are considerably [48,49] or only slightly elevated [46,54], while LH levels can be normal, mildly elevated [46], or low [8]. In some cases, the serum level of alpha subunit can be increased as well [55].

The relative amount of biologically active gonadotropins was postulated to be relatively low in many cases [56]. However, FSH biological activity was increased in two of six patients with gonadotropin-secreting adenomas (2.5 and 4.1, respectively; normal male range: 0.3–1.5) [8].

Serum inhibin concentration also did not discriminate between FGAs and NFAs [8] despite the fact that increased serum concentrations of inhibin B can be present in FGAs as a result of FSH stimulation [44]. It is likely that serum inhibin B level correlates with biological activity of FSH since inhibin B can be low despite increased serum FSH levels [48].

In most cases the tumor secretes only gonadotropins (either one, most frequently FSH, or both), but co-secretion of other pituitary hormones, namely Thyroid Stimulating Hormone (TSH) [57], Prolactin (PRL) [58], and Adrenocorticotropic Hormone (ACTH) [59], was reported.

### 5.3. Response to Dynamic Tests

Gonadotropins do not respond adequately to the administration of either gonadotropin-releasing factors [49,60,61] or gonadal steroids [49,61,62] as a proof of their at least partial autonomy.

In normal subjects, TRH has no effect on gonadotropin secretion, but in FGAs, serum gonadotropins levels are increased both basally and in response to TRH, and the increase was reported to be significantly greater than that obtained in controls or patients with NFAs [6,63]. Older reports suggested that the LH-β augmentation after TRH can better distinguish gonadotroph adenomas in men with clinically nonfunctioning PAs than basal FSH and α-SU levels [64]. However most recent studies revealed that the majority of NFAs release large amounts of β-LH, β-FSH, or both in response to TRH, limiting the usefulness of this test [65]. Measurement of α-SU after TRH test helps presurgical diagnosis [66,67] although the absence of α-SU in many tumors [67,68] is a confounding factor.

### 5.4. Sperm Count

Semen analysis can be normal even in the presence of markedly elevated basal FSH levels [49,69,70], again suggesting limited biological activity of FSH in some cases. However, increased sperm count (and increased serum testosterone) was described in a man with FGA that secreted FSH and LH [71]. Testicular biopsy in a man with elevated FSH levels revealed rare, apparently inactive Leydig cells, normal morphology of seminiferous tubules and Sertoli cells, and moderate hypospermatogenesis [62].

## 6. Children

FGAs are extremely rare in children. The majority of the cases reported were macroadenomas, and only two cases with assuming FGA microadenomas are described [72].

### 6.1. Clinical Presentation

Cases of central precocious puberty has been reported in girls [72,73] and boys [58,74] diagnosed at ages between 3 and 7 years old [58,73,74,75].

In postmenarcheal girls, amenorrhea can occur [76,77,78]. Ovarian enlargement is present in girls with FSH-secreting adenomas [79] and can be associated with nausea, abdominal distention, and pain as a result of bilateral multiple ovarian cysts [77]. Sometimes, an abdominal mass is palpable on clinical examination as an expression of giant ovarian cysts [76]. However, this type of presentation appears to be much rarer than in fertile years [80]. Testicular enlargement also occurs in affected boys [44].

### 6.2. Hormonal Tests

Elevated gonadotropin concentrations in children younger than 2 years may mimic mini-puberty [81], but no FGA has been reported so far in that age group. Nevertheless, precocious puberty caused by an LH-secreting adenoma was described as early as 3 years old [75].

In girls, elevated FSH and estradiol levels are described [73,77], sometimes together with supressed serum LH levels [76]. The diagnosis can be very difficult in the rare cases in which bilateral ovarectomy has been performed before reaching the correct diagnosis [78], and such a radical decision should only be considered after carefully examining the full range of differential diagnoses for isosexual central precocious puberty (inborn or acquired CNS lesions, genetic causes, etc.) [81].

In boys, either increased FSH and serum inhibin B levels with low LH concentrations [44] or increase in LH and testosterone [74] or clear elevation of serum levels of both gonadotropins and testosterone [58] are possible.

## 7. Management

Due to the rarity of the condition, there are no guidelines or large case series relevant to suggest the optimal management of FGAS. Transsphenoidal surgery is the first-line treatment and aims to resolve the hormonal syndrome and the pressure effect. Adjuvant radiotherapy has a role in the case of a postoperative remnant to prevent or treat recurrence. Medical treatment (dopaminergic agonists, somatostatin analogs, GnRH agonists and antagonists) has not been successful [82].

### 7.1. Surgery

Transsphenoidal adenomectomy (TSA) is the first-line treatment and may normalize endocrine function and resolve clinical symptoms (both those caused by tumor compression and tumor hypersecretion) [32,41,46,83].

In premenopausal women, successful cure is associated with resolution of ovarian hyperstimulation syndrome [28,83], decrease in ovarian size and cysts [32,79,84], restoration of regular menstrual cycles, and even normal pregnancy [19,30,32,34,85]. Bilateral ovarian cystectomy is not therapeutic, as cysts soon recur [36,86].

In males, successful TSA is followed by a marked decrease of previously elevated gonadotropin levels [46,49], a decrease in testicular volume [42], and clinical recovery [46]. Either hypogonadotropic hypogonadism needing replacement [51] or recovery of low testosterone levels can occur [87].

Good results are also reported after successful surgery in children, with resolution of precocious puberty symptoms and biochemical normalization [58,73,74,76].

However, local recurrence of FGAs or progression of tumor remnant can occur [88]. In one study, stable complete remission was described in six out of seven cases after surgery during a median of 10 months of follow-up [41]. No clear predictors of recurrence are currently available. The Ki-67 index is not a good predictor marker for progression or recurrence, while large tumor size and young age at diagnosis [88] have been proposed as prognostic markers in one study.

### 7.2. Medical Therapy

If surgery is unsuccessful or contraindicated, medical therapy can be offered to test for a positive response of FGAs.

#### 7.2.1. Oral Contraceptives

In two cases, diagnosis of FGA was made after cessation of oral contraceptives [9,16], but restarting them did not suppress FSH levels and ovarian stimulation in one of them [9]. Low-dose estrogen–progestin (LEP) has been administered in two females and suppressed the pituitary–gonadotropin axis, with subsequent decrease of ovarian volume only for the short term, as the ovarian swelling soon relapsed [22].

#### 7.2.2. Dopamine Agonists (DA)

Rare case reports suggested that particular sensitivity to DA can be present in some FGAs. Secretion of LH and α-SU decreased in response to a 4-hour dopamine infusion [89], and the same effect on both gonadotropins [69,90] and α-SU [91] has been reported as a result of prolonged DA treatment. In vivo, a small dose of bromocriptine (2.5–5 mg) significantly suppressed plasma hormone levels in some but not all patients [54,92].

Despite occasional old reports revealing rapid (3 days to 6 weeks) improvement in visual function, suggestive of a rapid tumor reduction effect [90,93] or more gradual reduction of tumor size with long-term bromocriptine administration [93,94], in most cases, no tumor shrinkage was noted, even in cases with hormonal response [54,69].

Short-term treatment (13 weeks) with cabergoline (0.5 mg weekly) resulted not only in normalization of previously elevated FSH and PRL levels but also in pregnancy [95]. However, in other cases, cabergoline was either ineffective [44] or only partially effective in normalizing increased hormonal concentrations [96].

In certain cases, both bromocriptine [40] and cabergoline [95,97] were reported effective in the management of ovarian hyperstimulation syndrome and resumed ovulation and fertility and, in some cases, have allowed pregnancy [37,40,95]. However, in other cases, DA administration worsened the clinical picture [98].

The high expression of dopaminergic 2 (D2) receptors and ERα might be associated with increased responsiveness rate and/or lower rate of relapse or progression, but the evidence so far is conflicting [17]. In a recent systematic review, DA treatment induced mild decrease of tumor secretion and/or decrease of ovarian diameter in 8 out of 18 patients [30].

Overall, DA are not considered useful for symptomatic or tumoral control, but since no predictors of a possible positive effect exist, a therapeutic trial should be considered in individual cases.

#### 7.2.3. Somatostatin Analogs (SSA)

The expression of somatostatin receptors type 2 and 3 has been described in gonadotroph tumors [99] but does not necessarily imply therapeutic efficacy, as proven by the case of increased tumor uptake at Indium 111-labeled octreotide scintigraphy in a FGA but complete lack of therapeutic effect of octreotide on either tumor secretion or size [100].

Similar to DA administration, in isolated cases, SSA treatment was clinically effective. SSA induced remission of OHSS caused by FGA in a young woman [101], improvement of visual field, and gradual decrease of tumor volume in one out of 2 FGAs (part of a larger series of PAs, mostly NFAs) [102]. Significant tumor secretion and size reduction was reported in a TSH-LH co-secreting PA [103]. However, in other case reports, no significant effect of SSA administration was noted on either tumor secretion or size [100], and the antisecretory effect was not constant [43].

Overall, the effect appears very limited, and similarly to DA, there is no possibility to discriminate potentially responsive cases. A rapid octreotide test was advocated to check for potential clinical efficacy, but the significant antisecretory effect on both LH and alpha-subunit in a woman with FGA after acute octreotide administration was not replicated by long-term lanreotide treatment [104].

Combined therapy with SSA-DA was associated with transient symptomatic improvement, partial hormonal effect, and slight reduction of tumor size in one case [48] or persistent reduction of tumor secretion for 6 months [53]. Cases treated with pasireotide, a multi-receptor-targeted SSA, are not reported.

#### 7.2.4. GnRH Agonists/Antagonists

Antisecretory effect of GnRH antagonist was reported in some [98,105,106] but not all studies [35]. In one study, the administration of Nal-Glu GnRH to patients with FSH-secreting adenomas produced a significant decrease in the serum FSH concentration in two out of four patients and normalized the FSH level in one case [106]. In another study, a single administration in seven patients with FSH-secreting adenomas after unsuccessful surgery produced a slight, but a significant fall in FSH levels in all cases as early as 12 h after injection yet wide interindividual variability of response was described [105]. However, in a woman with FSH-secreting adenoma, FSH and alpha-subunit did not decrease after administration of a GnRH [35]. Similarly, the administration of GnRH antagonists was found to be effective for the short-term treatment of ovarian hyperstimulation syndrome in some reports [98] but not in others [38].

FGA cells in culture demonstrate variable stimulatory responses to acute GnRH administration, while treatment with GnRH agonist decreased tumor secretion (as a result of the desensitizing effect of medication) in only one out of three tumors [107]. Similarly, in vivo, one patient with FGA failed to show a desensitization effect as expected [43]. Increase in FSH in response to a GnRH agonist and subsequent worsening of the clinical picture is possible [34] as well as an increase in tumor size with the risk of pituitary apoplexy [108,109]. Nevertheless, the good antisecretory response to the acute administration of GnRH agonist leuprolide failed to predict a response to chronic treatment [106].

Longer treatment (3–12 months) with GnRH analogs induced only antisecretory effect and no tumor shrinkage [110,111] with the exception of occasional reports of tumor shrinkage [71]. Moreover, in a recent systematic review, GnRH agonists/antagonists were dispensed in eleven patients, and partial symptomatic and biochemical effect occurred in two but aggravation (including tumor increase) in four [30].

The generally accepted view is that medical therapies are partially or totally ineffective in many cases, especially with respect to antitumor effect; despite clinical or hormonal improvement in some cases [30], this approach should not be considered the first option.

### 7.3. Radiotherapy (RT)

RT has been used both for tumor regrowth [30,41,69] or to prevent remnant progression after partial resection [60,71,96], but there is a lack of long-term follow-up data.

The decision to offer adjuvant radiotherapy should be taken on an individual basis in the setting of a multidisciplinary team. One reported case proved that small sub-centimetric remnant can remain clinically silent for many years [86], suggesting that it might be better to postpone the decision until clear tumor progression is proven. In another case, after repeated surgery, the tumor remnant remained stable during 6 years of follow-up [41]. Older, conventional RT types were not always successful in terms of biochemical control [60,69] even with long (12 years) follow-up [60]. From the experience with other, more common type of PAs, it is probable that better results can be expected after conventionally fractionated RT and even more so after radiosurgery, which should be considered the first-choice radiotherapeutic modality [112,113]. Occasional case reports described tumor control at least 8 years after stereotactic radiotherapy offered for postoperative tumor recurrence [41].

## 8. Long-Term Outcome

The long-term outcome of FGAs is unknown since most reports do not have a prolonged follow-up. Natural history of FGAs can sometimes be characterized by spontaneous fluctuations in both the laboratory tests and clinical expression, as in the case of a 40-year-old woman with marked fluctuations of estradiol and ovarian size over 1 year of follow-up [114].

Gonadotropic adenomas have a high risk of recurrence/progression, which makes surveillance mandatory. The antero-posterior AP diameter is the strongest predictor of tumor recurrence/progression, and each 5 mm increase in AP diameter doubles the hazard ratio of recurrence/progression [88]. Previous studies show that relapse is more likely in “large” non-functional tumors [115].

A review of 50 published cases of FGAs in women of reproductive age revealed gross total resection in 12/25 cases and tumor recurrence in 5 of them over a mean follow-up of 25 months [30]. Recurrences can be managed by reoperation [30,36], radiotherapy [30,101], or medical treatment, alone or in addition to radiotherapy [53,101].

In an even more recent systematic review of 65 premenopausal women with FGAs, 63 of which (96.9%) were operated, 77.8% of the operated cases achieved complete tumor excision and disappearance of symptoms, but the follow-up was too short to draw valuable conclusions about the recurrence risk. The others received either radiotherapy or repeated surgery. In two cases, observation was chosen, while in another, ovarian surgery was decided with no additional pituitary-directed therapy [21].

Although tumor may not recur for as long as 12 years [24], long-term follow-up (clinical, biochemical, and imaging) is mandatory.

The fertility prospects are good with successful treatment: in the retrospective review mentioned, 12 out of 14 cases with full pregnancy data conceived [30]. If spontaneous pregnancy cannot be achieved after multimodal treatment, in vitro fertilization (IVF) with embryo transfer can prove successful [116] even in cases with persistent tumor hypersecretion [24].

A very rare possibility is that of pituitary carcinoma diagnosed years after an initial diagnosis of FGAs in the context of multiple endocrine neoplasia (MEN) 1 syndrome. The initial tumor recurred repeatedly despite iterative surgery and radiotherapy, and eventually, a brain metastasis with features of de-differentiation (staining only for steroidogenesis factor-1 and not for gonadotropin subunits) was diagnosed [117].

Recently, Principe et al. showed that gonadotroph pituitary neuroendocrine tumors (PitNETs) demonstrate an increased CD68+ macrophage environment compared to somatotroph, lactotroph, and corticotroph PitNETs. A significant correlation between CD 68+ and CD 163+ infiltrating macrophages and the invasive profile of gonadotroph tumors has been revealed [118]. This finding implies that macrophage-targeted immunotherapies could be a therapeutic target to limit the progression of gonadotroph PitNETs. Temozolomide (TMZ) alone or in combination with 5 Fluorouracil (5FU) or with RT can be an effective treatment of aggressive PAs [119].

## 9. Conclusions

FGAs represent a rare group of tumors. They provoke distinct clinical manifestations in premenopausal females (mainly menstrual irregularity and ovarian hyperstimulation syndrome, testicular enlargement or hypogonadism in males, and rarely isosexual precocious puberty in children. Their pathogenesis remains unknown. A high suspicion index is needed for early diagnosis and targeted treatment. Transsphenoidal surgery is the mainstay treatment, while medical treatment (dopaminergic agonists, somatostatin analogs, GnRH agonists and antagonists) have given disappointing results. Irradiation may be discussed in case of a postoperative remnant to prevent or treat recurrence.

## Data Availability

Not applicable.

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
