# Peer review of "Updating the Landscape for Functioning Gonadotroph Tumors"

_medicina, 2022, doi:10.3390/medicina58081071_

Round 1
Reviewer 1 Report
This manuscript by Georgia Ntali and Cristina Capatina, submitted to Medicina, entitled “Updating the landscape for functioning gonadotroph tumors” is a comprehensive and well updated review on functional gonadotroph adenomas.
1. Stating as a matter of start that “Functioning gonadotropin adenomas (FGAs) have been unexplored until today” seems a too strong statement. It would be more appropriate to underline the rarity of these tumors and why the overwhelming majority of gonadotroph adenomas, one of the most frequent form of pituitary tumors, are clinically nonfunctioning.
2. The review should highlight what is specific to functioning gonadotroph tumors rather than covering aspects that are well described for gonadotroph adenomas in general, like tumors mass effects or signs related to tumor-related hypopituitarism.
3. As mentioned the exact prevalence of FGAs is unknown. A single study is cited relating to this issue; it is likely overestimating this prevalence.
4. Apart from the issue of glycosylation, other mechanisms such as the need for both alpha and beta subunits or for a pulsatile pattern of secretion should be further discussed, accounting for the nonfunctioning nature of most gonadotroph adenomas.
5. Lines 118-121 are unclear: in postmenopausal women, low levels of LH or discrepancy between FSH and LH may be an indication of gonadotroph adenoma, but not specifically of a functioning one.
Author Response
Dear Reviewer,
Thank you very much for your remarks which have helped to improve to our manuscript. We have now revised it according to them.
Comment 1) Stating as a matter of start that “Functioning gonadotropin adenomas (FGAs) have been unexplored until today” seems a too strong statement. It would be more appropriate to underline the rarity of these tumors and why the overwhelming majority of gonadotroph adenomas, one of the most frequent form of pituitary tumors, are clinically nonfunctioning.
Answer to comment 1: We have now changed the sentence to Functioning gonadotropin adenomas (FGAs) are rare tumors as the overwhelming majority of gonadotroph tumors are clinically silent. Gonadotroph tumors are poorly differentiated and produce and secrete hormones inefficiently but in exceptional cases they cause clinical syndromes due to hypersecretion of intact gonadotropins.
We also changed that statement in the conclusions.
Comment 2. The review should highlight what is specific to functioning gonadotroph tumors rather than covering aspects that are well described for gonadotroph adenomas in general, like tumors mass effects or signs related to tumor-related hypopituitarism.
Answer to comment 2: The sentence that refered to mass effect and hypopituitarism has been deleted and replaced by ….Transsphenoidal surgery and removal of tumor, reduces hormonal hypersecretion, improves endocrine dysfunction and provides tissue for further analysis
Comment 3. As mentioned the exact prevalence of FGAs is unknown. A single study is cited relating to this issue; it is likely overestimating this prevalence.
Answer to comment 3. As suggested a comment that the estimated prevalence by Ceccato et al can be an overestimation has been added.
Interestingly in a retrospective study of 171 premenopausal women operated for a macroadenoma Caretto et al detected 3 cases with clinical and biochemical characteristics of OHSS and estimated that the prevalence could reach 8.1% in the gonadotroph adenoma group and 2.9% in the NFA group [4], nevertheless this has not been validated in other series and could represent an overestimation
Comment 4. Apart from the issue of glycosylation, other mechanisms such as the need for both alpha and beta subunits or for a pulsatile pattern of secretion should be further discussed, accounting for the nonfunctioning nature of most gonadotroph adenomas.
Answer to comment 4. Other mechanisms that could explain the clinical syndromes associated with FGAs have been added.
Alteration of the pulsatile secretion of gonadotropins and a slight but constant release of FSH could stimulate recruitment of multiple dominant follicles and release of high serum estradiol similarly to ovarian stimulation with exogenous FSH when administered for fertility treatment. Modifications of their heterodimer of alpha and beta chain can potentially lead to an increased bioactivity of FSH levels.
Comment 5. Lines 118-121 are unclear: in postmenopausal women, low levels of LH or discrepancy between FSH and LH may be an indication of gonadotroph adenoma, but not specifically of a functioning one.
Answer to comment 5. The sentence in line 126 has been changed as suggested in order to make clear that diagnosis of an FGA during menopause can not be ascertained by FSH and LH levels.
As gonadotropin increase is secondary to menopause, low levels of LH levels or discrepancy of FSH and LH could be an indication of a gonadotroph adenoma but not specifically of a functioning one.

Reviewer 2 Report
The present literature review about the functioning gonadotroph tumours is well written and data reported quite interesting; in my opinion it could be of real interest for Medicina journal readership. On the other hand few minor problems are encountered as follow:
1) I wonder if the exact paper title is "functioning gonadotroph tumors" or "updating the landscape for functioning gonadotroph tumors"? Please explain/amend.
2) It is not clear what kind of literature review the authors have performed, is it a narrative, a PRISMA or what else? Please specify and add few lines in the methods section about it and if necessary a table summarizing all reviewed/included papers.
3) All paper sections are too long especially the "pathogenesis-pathology" thus all should be shorten, this is supposed to be an update paper not a book chapter (please amend).
Author Response
Dear Reviewer
Thank you very much for your remarks which have helped to improve to our manuscript. We have now revised our manuscript according to them.
Responses to Reviewer
Comment 1) I wonder if the exact paper title is "functioning gonadotroph tumors" or "updating the landscape for functioning gonadotroph tumors"? Please explain/amend.
Answer to comment 1: The exact paper title is updating the landscape for functioning gonadotroph tumors.
Comment 2) It is not clear what kind of literature review the authors have performed, is it a narrative, a PRISMA or what else? Please specify and add few lines in the methods section about it and if necessary a table summarizing all reviewed/included papers.
Answer to comment 2. This is a narrative review based on English literature review in pubmed.
A paragraph Methodology was added after introduction in order to specify details.
We performed a narrative review, based on relevant articles written in English from a Pubmed search, using the following search terms: "functioning gonadotroph adenoma", "ovarian hyperstimulation syndrome", "testicular enlargement", and "precocious puberty". Most papers were case reports or small case series.
We appreciated that a table summarizing all included papers would not be of a value as the papers have been extensively reviewed in the relevant sections.
3) All paper sections are too long especially the "pathogenesis-pathology" thus all should be shorten, this is supposed to be an update paper not a book chapter (please amend).
Answer: Shortening of the paper has been done in all sections including pathogenesis.
